# Aliphatic-Bridged Early Lanthanide Metal–Organic Frameworks: Topological Polymorphism and Excitation-Dependent Luminescence

Pavel A. Demakov ⬤, Alexey A. Ryadun and Vladimir P. Fedin *⬤

Nikolaev Institute of Inorganic Chemistry SB RAS, 3 Lavrentiev Ave., 630090 Novosibirsk, Russia
* Correspondence: cluster@niic.nsc.ru

**Abstract:** Six new three-dimensional metal–organic frameworks based on early lanthanide(III) cations and *trans*-1,4-cyclohexanedicarboxylic acid ($H_2$chdc) were obtained. Their crystal structures were determined by single-crystal X-ray diffraction analysis. The structure of $[La_2(H_2O)_4(chdc)_3]\cdot2DMF\cdot H_2O$ (**1**; DMF = N,N-dimethylformamide) contains one-dimensional infinite La(III)-carboxylate chains interconnected by cyclohexane moieties to form a highly porous polymeric lattice with 30% solvent accessible volume. Compounds $[Ln_2(phen)_2(chdc)_3]\cdot0.75DMF$ (**$2_{Ln}$**; $Ln^{3+}$ = $Ce^{3+}$, $Pr^{3+}$, $Nd^{3+}$ and $Sm^{3+}$; phen = 1,10-phenanthroline) are based on binuclear carboxylate building blocks, which are decorated by chelate phenanthroline ligands and interconnected by cyclohexane moieties to form more dense isostructural coordination frameworks with primitive cubic *pcu* topology. Compound $[Nd_2(phen)_2(chdc)_3]\cdot2DMF\cdot0.67H_2O$ (**3**) is based on secondary building units similar to **$2_{Ln}$** and contains a coordination lattice isomeric to **$2_{Ln}$** with a rare hexagonal helical *snz* topology. Thermal stability and luminescent properties were investigated. For **$2_{Sm}$**, a strong and nonmonotonous dependence of the luminescence color on the variation of excitation wavelength was revealed, changing its emission from pinkish red at $\lambda_{ex}$ = 340 nm to white at $\lambda_{ex}$ = 400 nm, and then to yellow at lower excitation energies. Such nonlinear behavior was rationalized in terms of the contribution of several different luminescence mechanisms. Thus, **$2_{Sm}$** is a rather rare example of a highly tunable monometallic lanthanide-based luminophore with possible applications in light-emitting devices and optical data processing.

**Keywords:** coordination polymers; metal–organic frameworks; polymorphism; rare earth elements; luminescence

## 1. Introduction

Metal–organic frameworks (MOFs) represent an emerging class of hybrid inorganic–organic materials. A design of diverse functional properties, e.g., catalytic, optical, adsorption and magnetic, is possible in MOFs by carefully choosing metal building blocks and organic bridges [1–6]. The effects of reversible structural transitions (breathing) [7–10] and topological polymorphisms [11–13] inherent to metal–organic coordination polymers are of great interest due to the corresponding ability of fine tuning the porosity, crystal packing of functional centers and strength of host–guest interactions. The aliphatic-backboned ligands bearing their own enhanced structural mobility and specific optical, thermal and adsorption properties [14–17] deserve special attention in MOF chemistry. In particular, the optical transparency of such ligands gives us the perfect ability to investigate the photophysical properties of photoactive coordination nodes or guest molecules incorporated into the aliphatic-spread polymeric network [18,19].

Lanthanide-based metal–organic frameworks are extensively studied owing to their remarkable luminescence [20–23] originating from the emission of metal cation when organic ligands and guest molecules with conjugated π-systems often act as photosensitizers for self-emitting $Ln^{3+}$. In particular, combining lanthanide(III) ions bearing different emission

colors and/or using highly emissive organic guests or ligands within $Ln^{3+}$-based MOFs is a convenient route for the synthesis of single-phase white-light emitters suitable for real applications [24–29]. High coordination numbers of Ln(III) ions and the corresponding abundant structural variability of polymeric coordination lattices [30–32] push ahead their synthesis and comprehensive investigations. $Ln^{3+}$-based MOFs are also characterized by higher thermal and hydrolytic stabilities compared to more common frameworks based on divalent transition metal cations [33–35], which makes them suitable for applications in sensor devices, light emitters and bioimaging [36–40].

Previously, we have reported a series of metal–organic frameworks based on 1,10-phenanthroline (phen) and *trans*-1,4-cyclohexanedicarboxylic acid ($H_2$chdc) with the formulas [$Ln_2$(phen)$_2$(chdc)$_3$]·0.5DMF ($Ln^{3+}$ = $Eu^{3+}$, $Gd^{3+}$, $Tb^{3+}$, $Y^{3+}$) [28,41], for which a strong and excitation wavelength-dependent luminescence with quantum yields of up to 63% was observed. The origin of luminescence varied from metal-centered antenna-sensitized emission for $Eu^{3+}$ and $Tb^{3+}$ compounds to the ligand-centered emission for the compounds with non-emissive $Gd^{3+}$ or $Y^{3+}$ cations. Tuning the metal coordination environment by nitrate, chloride or carboxylate in a series of structurally kindred Gd(III) metal–organic frameworks with an optically transparent *trans*-1,4-cyclohexahedicarboxylate bridge allowed us to understand the impact of the coordinated X anion on the ligand-centered luminescence in photoactive {$Gd_2$(phen)$_2$(X)$_2$(OOCR)$_4$} moiety. Owing to the intensive and highly tunable luminescence of several MOFs of the [$Ln_2$(phen)$_2$(chdc)$_3$] family, we further focused on the synthesis and investigation of [$Ln_2$(phen)$_2$(chdc)$_3$]-type metal–organic frameworks with early lanthanides based on an isostructural binuclear carboxylate building block (Figure 1). Early $Ln^{3+}$ cations are also known to have unique emission characteristics. As a result, this work reports the successful isolation and characterization of six new MOFs constructed by early lanthanides and $H_2$chdc with the corresponding crystallographic formulae [$La_2$($H_2$O)$_4$(chdc)$_3$]·2DMF·$H_2$O (**1**; DMF = N,N-dimethylformamide), [$Ln_2$(phen)$_2$(chdc)$_3$]·0.75DMF (**2$_{Ln}$**; $Ln^{3+}$ = $Ce^{3+}$, $Pr^{3+}$, $Nd^{3+}$ and $Sm^{3+}$) and [$Nd_2$(phen)$_2$(chdc)$_3$]·2DMF·0.67$H_2$O (**3**). The numbers **1**–**3** represent different structural types inherent to the compounds obtained in close conditions. A non-photosensitized luminescence was revealed for **2$_{Nd}$**, while **2$_{Sm}$** demonstrated strongly excitation-dependent color of emission, which is rather unusual for monometallic Ln(III) metal–organic frameworks.

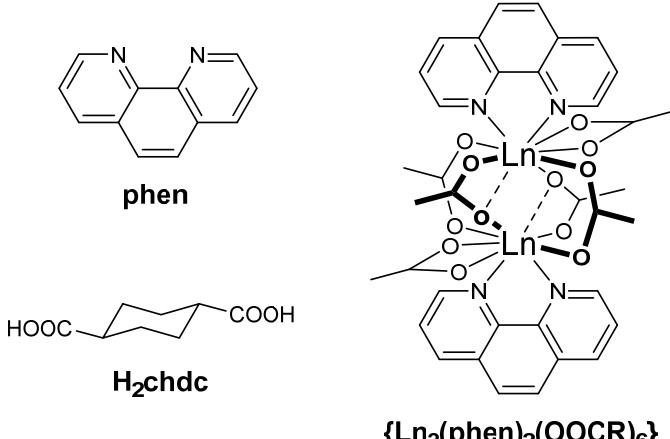

**Figure 1.** Structural formulae of the used ligands and secondary building unit in **2$_{Ln}$**.

## 2. Results and Discussion

### 2.1. Synthesis

Single crystals of compound [$La_2$($H_2$O)$_4$(chdc)$_3$]·2DMF·$H_2$O (**1**) were obtained by the reaction between lanthanum(III) chloride, *trans*-1,4-cyclohexanedicarboxylic acid ($H_2$chdc) and 1,10-phenanthroline (phen) in a mixture of N,N-dimethylformamide (DMF) and water at 80 °C. Phenanthroline is not comprised in **1**; however, carrying the reaction without

phen in similar conditions resulted in the formation of an unidentified crystalline phase. The role of 1,10-phenathroline in the synthesis could be suggested in terms of its slight basicity [42,43], promoting the deprotonation of weak organic acid $H_2$chdc. The PXRD pattern of the filtered sample of **1′** shows its poor crystallinity, which might be attributed to the mobility of highly porous coordination framework based on conformationally flexible chdc ligand after removal of the mother liquor and subsequent desolvation of **1** (see pages 10–12 in ESI). Chemical analysis of **1′** samples after the storage (see Section 3.3 in the experimental) confirms the substantial decrease in guest molecule content, compared to the crystallographic formula of **1**.

Compounds $[Ln_2(phen)_2(chdc)_3]\cdot0.75DMF$ ($2_{Ln}$; $Ln^{3+} = Ce^{3+}$, $Pr^{3+}$, $Nd^{3+}$ and $Sm^{3+}$) were synthesized in conditions quite similar to **1,** except for changing lanthanum(III) chloride to the corresponding amounts of other lanthanide(III) salts and some variation in solvent DMF: $H_2O$ ratio. This synthetic method was optimized to avoid the formation of unidentified crystalline phases. PXRD patterns of $2_{Ln}$ (Figures S1–S4) correspond well to the theoretical ones, indicating their successful synthesis. Unlike **1**, crystal structures of $2_{Ln}$ contain a phen molecule being coordinated to each Ln(III) ion. Such difference between La-based **1** and compounds $2_{Ln}$ is apparently attributed to the widely known effect on the decrease in the atomic and ionic radii in the lanthanide row with increasing atomic number (called lanthanide contraction), which results in the possible chemical and structural dissimilarity [30,44,45] of the forming coordination compounds; the latter is more comprehensively discussed in the next part.

Single crystals of $[Nd_2(phen)_2(chdc)_3]\cdot2DMF\cdot0.67H_2O$ (**3**) were obtained in low yield by the reaction of neodymium(III) chloride, $H_2$chdc and phen in DMF at 110 °C. The formation of **3** appears to compete strongly to the crystallization of $2_{Nd}$. Time screening at 110 °C revealed that **3** crystallizes as a major product only during several hours of heating (Figure 2), while increasing the synthesis time up to two days carrying all other similar conditions leads to pure $2_{Nd}$ precipitation. In this regard, **3** can be recognized as a metastable kinetic reaction product. Temperature screening at a short synthesis time showed that **3** forms as only a minor admixture to $2_{Nd}$ at 120 °C, while no precipitation was observed at 100 °C and 130 °C. Surprisingly, no formation of **3**-like phases was observed in similar screening syntheses using Pr(III) and Sm(III) (Figures S5 and S6), which are the nearest available neighbors of neodymium in the lanthanide row, which puts Nd-based **3,** topologically isomeric to the $2_{Nd}$ coordination framework, in a completely unique place.

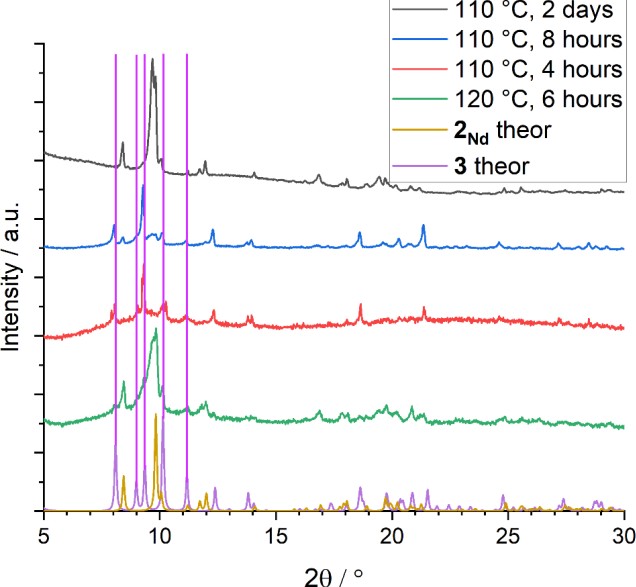

**Figure 2.** PXRD patterns for $2_{Nd}$/**3** temperature- and time-screening syntheses. Magenta lines show positions for lowest-angle reflections from the theoretical **3** structure.

### 2.2. Crystal Structure Description

A suitable crystal of **1** was taken directly from the mother liquor before the filtration for performing single-crystal XRD. Compound **1** crystallizes in the triclinic symmetry with *P*–1 space group. The asymmetric unit of **1** includes one La atom. La(III) is coordinated by two O atoms of two $\kappa^2$- carboxylic groups, four O atoms of two chelated $\kappa^2,\kappa^1$- carboxylic groups, two O atoms of two nonchelated $\kappa^2,\kappa^1$- carboxylic groups and two O atoms of two water molecules. Therefore, the La atom adopts the coordination number 10. La–O(COO) bond lengths range from 2.524(3) Å to 2.771(3) Å and La–O(H$_2$O) bond lengths are 2.525(3) Å and 2.543(3) Å. La atoms are interconnected by $\kappa^2,\kappa^1$- carboxylate bridges forming one-dimensional {-La(OH$_2$)$_2$(OOCR)$_3$-}$_n$ chains (Figure 3a), which act as building units in the coordination framework. These chains are interconnected by cyclohexane moieties to form a three-dimensional polymeric lattice, in which chdc ligands present in two different ($\kappa^2,\kappa^1;\kappa^2,\kappa^1$) and ($\kappa^2;\kappa^2$) coordination modes (Figure S7), situated along the *b* and *c* axes, respectively. The coordination framework in **1** contains the system of one-dimensional channels running in two perpendicular directions. The larger channels paved by cyclohexane rings (Figure 3b) are situated along the *a* crystallographic axis and are ca. 4 × 4 Å$^2$ in size (Figure S8a). The smaller channels paved by cyclohexane rings and coordinated water molecules (Figure 3c) are situated along the *b* crystallographic axis and are ca. 3 × 4 Å$^2$ in size (Figure S8b). These two types of channels intersect and the total solvent accessible volume in **1** reaches 30%. The channels are fully occupied by guest DMF and water molecules.

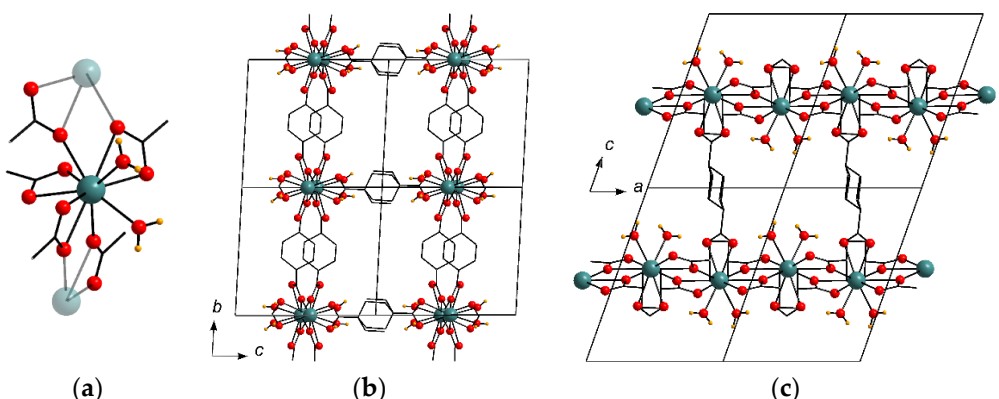

|     |     |     |
| --- | --- | --- |
| **(a)** | **(b)** | **(c)** |

**Figure 3.** La(III) environment in **1**. The neighboring La(III) ions are shown as transparent. O atoms are red, H atoms are orange (**a**). View along different crystallographic axes in **1**: *a* (**b**) and *b* (**c**). Guest molecules and H atoms of chdc ligand are not shown. Only one possible position of disordered ($\kappa^2;\kappa^2$)-chdc cyclohexane ring is shown.

Compounds **2$_{Ce}$**, **2$_{Pr}$**, **2$_{Nd}$** and **2$_{Sm}$** were obtained under similar conditions and are isostructural. Therefore, only the **2$_{Nd}$** structure will be described in detail as a representing example, which is also may be conveniently compared to the **3$_{Nd}$** structure described below. **2$_{Nd}$** crystallizes in the monoclinic symmetry with the *P*2$_1$/*n* space group. Nd(III) is coordinated by two O atoms of the $\kappa^2$- carboxylic group, two O atoms of the chelated $\kappa^2,\kappa^1$- carboxylic group, one O atom of the nonchelated $\kappa^2,\kappa^1$- carboxylic group, two O atoms of two $\kappa^1,\kappa^1$- carboxylic groups and two N atoms of the phenanthroline chelate molecule. Therefore, the Nd atom adopts the coordination number 9. The bond lengths of the metal coordination environment in **2$_{Ln}$** are presented in Table S3. Two symmetry-equivalent Nd atoms are coupled into binuclear {Nd$_2$(phen)$_2$(OOCR)$_6$} building blocks (Figure 4a) acting as six connected nodes in the coordination framework. Importantly, the lanthanum-based **1** was obtained in conditions similar to **2$_{Ln}$**, including the addition of phenanthroline, despite the metal salt precursor. Therefore, a lanthanide contraction leads not only to the decrease in the metal coordination number from 10 (**1**) to 9 (**2$_{Ln}$**), but also to the coordination of phen instead of two water molecules and to the change in the structural type of the secondary

building unit from infinite 1D chains (**1**) to separate binuclear carboxylate blocks (**2Ln**). These blocks are interconnected by cyclohexane moieties, forming a three-dimensional polymeric lattice (Figure 4b) with primitive cubic (*pcu*) topology (Figure 4c). As it was found by the PLATON routine [46], the voids in **2Nd** are represented by small isolated cages with only 6% total solvent accessible volume. These cages are filled by partially occupied guest DMF positions with 0.75DMF content per formula unit.

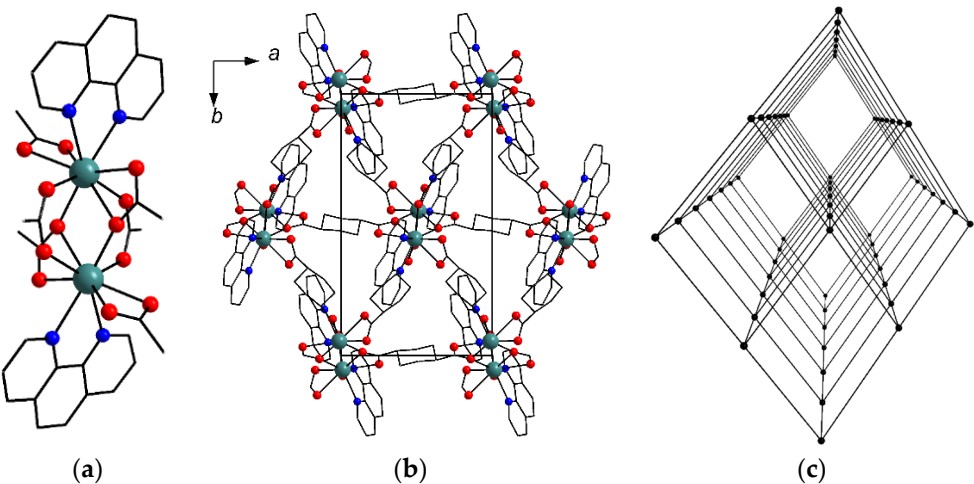

(**a**)  (**b**)  (**c**)

**Figure 4.** Binuclear block {$Nd_2$(phen)$_2$(OOCR)$_6$} in the structure of **2Nd** (**a**) and view along *c* crystallographic axis (**b**). O atoms are red, N atoms are blue, H atoms and guest molecules are not shown. Topological representation of the coordination lattice in **2Nd** (**c**). Binuclear block {$Nd_2$(phen)$_2$(OOCR)$_6$} is shown as a black node.

Compound **3** crystallizes in the trigonal symmetry with the *R*–3 space group. The Nd(III) coordination environment and structure of the binuclear {$Nd_2$(phen)$_2$(OOCR)$_6$} carboxylate block (Figure 5a) are similar to those in **2Nd**. However, the packing of such building units is different, which leads to the helical *snz* topology (Figure 5c) of a coordination lattice, rarely occurring in MOFs [47,48]. The framework in **3** contains one-dimensional channels running along *c* crystallographic axis (Figure 5b). These channels are paved by phenanthroline molecules and are ca. $4 \times 4$ Å$^2$ in size. The total solvent accessible volume in **3** is more than in **2Nd** and reaches 8%. The channels are fully occupied by guest DMF and water molecules.

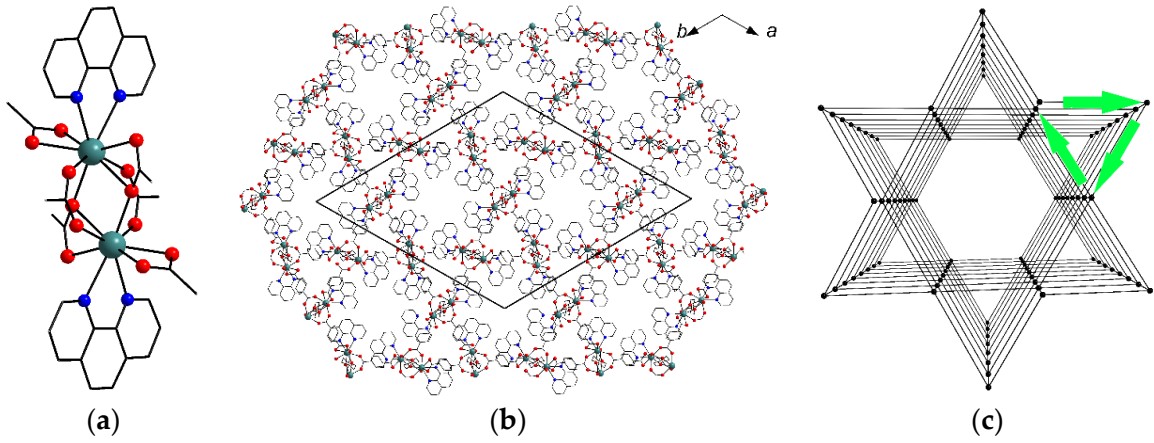

(**a**)  (**b**)  (**c**)

**Figure 5.** Binuclear block {$Nd_2$(phen)$_2$(OOCR)$_6$} in the structure of **3** (**a**) and view along *c* crystallographic axis (**b**). Atom colors correspond to Figure 4; guest molecules are not shown. Topological representation of the coordination lattice in **3** (**c**). Binuclear block {$Nd_2$(phen)$_2$(OOCR)$_6$} is shown as a black node. Green arrows show a helical structure of trilateral channels in the net.

### 2.3. Characterization and Thermal Properties

The crystal structure of **1** was found to be unstable after filtration. The PXRD pattern of the filtered sample shown in Figure S9a demonstrates a significant decrease in its crystallinity. The interpretation of chemical and physicochemical analyses performed for this sample, which is denoted as **1′**, is provided in pages 10–12 of the Supplementary Information file.

Infrared spectra (Figure S11) of the samples **2$_{Ln}$** correspond well to their compositions firstly determined by the single-crystal XRD. The spectra of **2$_{Ln}$** contain typical cyclohexane ring and DMF methyl group $C(sp^3)$–H vibrations, DMF C=O vibrations and COO-group antisymmetric and symmetric stretchings. A presence of weak coupled bands at 3075–3060 cm$^{-1}$ corresponding to the valence vibrations of $C(sp^2)$–H bonds, as well as strong bands at ca. 1583, 1534 and 850 cm$^{-1}$, belonging to the phenanthroline oscillations [49,50], prove the successful incorporation of the phen ligand into coordination frameworks **2$_{Ln}$** instead of **1**, fully consisting with their crystal structures. Elemental CHN and thermogravimetric data (Figure 6) also confirm the chemical nature and guest composition of the synthesized MOFs **2$_{Ln}$**. According to TGA, slow solvent loss occurs in the range of 250–450 °C; such a high temperature of guest DMF and water release is apparently attributed to their isolation into very narrow-windowed pores (see above). Decomposition of the coordination frameworks in **2$_{Ln}$** occurs at 470–510 °C, according to the temperatures of weight loss rate local maxima. Weights of final residues at 600 °C exceed the calculated weights of lanthanide oxides (Table S4), which indicates a presence of a considerable amount of carbon admixture in the thermolysis products of **2$_{Ln}$**, due to the incomplete evaporation of organic moieties. Such thermal stability fits typical values for the phenanthroline-coordinated carboxylate MOFs of early lanthanides [51–54].

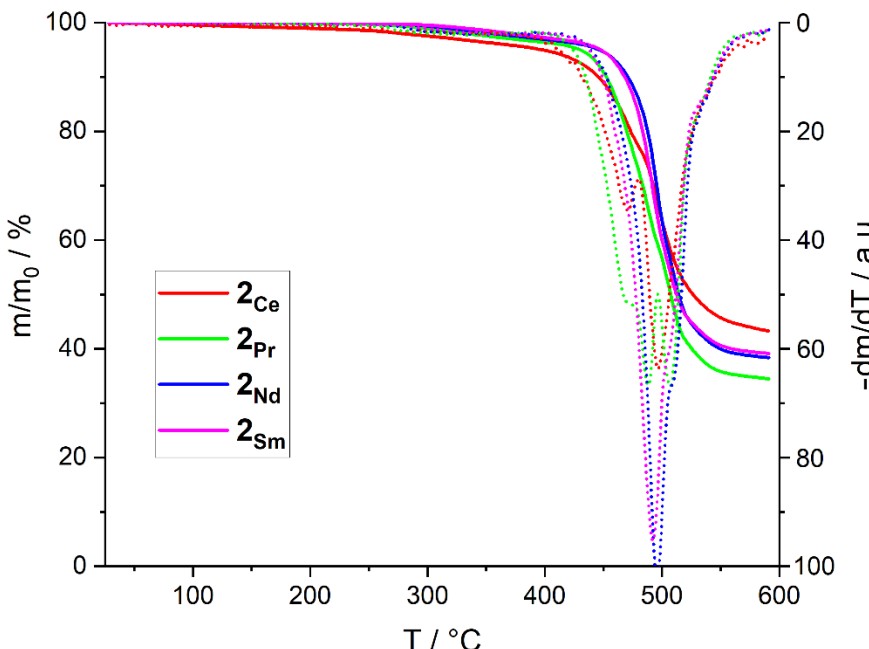

**Figure 6.** TG plots for compounds **1′**, **2$_{Ln}$** (solid) and relative dm/dT (dashed).

### 2.4. Luminescent Properties

As it was described in the introduction, a series of metal–organic frameworks with the formulae [Ln$_2$(phen)$_2$(chdc)$_3$]·0.5DMF (Ln$^{3+}$ = Eu$^{3+}$, Gd$^{3+}$, Tb$^{3+}$, Y$^{3+}$), in which the coordination lattice is isostructural to **2$_{Ln}$**, was reported by us previously. Highly effective and tunable luminescence was revealed for such series. Since that, the investigation of luminescent properties of **2$_{Ln}$**, especially for **2$_{Sm}$** containing a moderately emissive Sm$^{3+}$ cation, has been of great interest. The luminescent measurements revealed no emission

for $2_{Ce}$ and $2_{Pr}$, possibly due to the strong paramagnetic quenching of the corresponding cations. For $2_{Nd}$, a weak infrared emission with $\lambda_{em}$ = 1102 nm was observed under 804 nm excitation (Figure S12). This emission band is attributed to the $^4F_{3/2} \rightarrow {}^4I_{11/2}$ transition in $Nd^{3+}$ [55–57] and the excitation band might be attributed to the $^4F_{9/2} \rightarrow {}^4I_{11/2}$ transition [58,59] or to $^4F_{5/2}, {}^4H_{9/2} \rightarrow {}^4I_{9/2}$ transitions [60], indicating no apparent sensitization of $Nd^{3+}$ luminescence by phenanthroline.

Solid-state luminescent properties of $2_{Sm}$ were also investigated. The excitation spectrum is shown in Figure S13. The emission spectrum at $\lambda_{ex}$ = 340 nm (Figure 7a) contains a wide several-moded band in the UV-violet region, apparently attributed to the phenanthroline ligand-centered emission, and a series of strong bands at $\lambda_{em}$ = 564 nm, 598 nm and 646 nm, typical to the $Sm^{3+}$-centered emission and corresponding to the series of $^4G_{5/2} \rightarrow {}^6H_J$ (J = 5/2, 7/2, 9/2, respectively) transitions. The integral emission color of $2_{Sm}$ at $\lambda_{ex}$ = 340 nm is pinkish-red (Figure 7b), as is seen from the CIE 1931 chromaticity diagram. $Sm^{3+}$ is known to be a mild energy acceptor in photosensitization processes, compared to $Eu^{3+}$, representing the strongest emitter among Ln(III) cations. Therefore, the emission of $Sm^{3+}$ in its coordination complexes often competes to the intraligand emission of the coordinated photoactive species. In different MOF cases, $Sm^{3+}$ emission may be either dominant [61–63] or almost silent [64–66], the latter indicating ineffective energy transfer to the metal ion from organic antenna. Moreover, the emission intensities of both $Sm^{3+}$ and ligand components can be comparable to each other [67,68], which could unveil a route for a pronounced color tuning in a simple single-lanthanide system, which is preferrable for real applications, although there are some reported nice examples of effective emission color tuning in Sm-MOFs by codoping with other lanthanides [64,69,70]. Since the emission of $2_{Sm}$ belongs to the third of the types described above, and owing to the $\lambda_{ex}$ dependence of emission color previously reported by us for close (Y/Eu/Tb)-phen-chdc systems [28], we decided to further study the luminescence of $2_{Sm}$ in a wide range of excitation wavelengths.

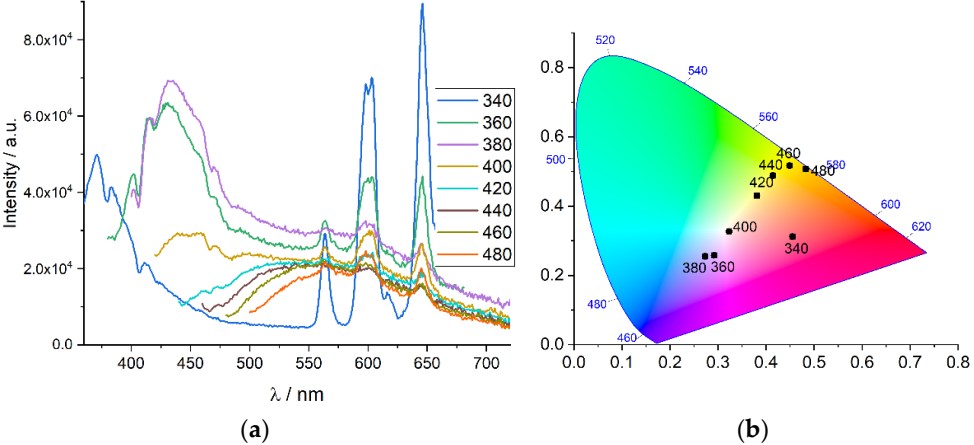

(a)                                                                                    (b)

**Figure 7.** Emission spectra for $2_{Sm}$ at different excitation wavelengths (**a**). CIE 1931 chromaticity diagram for $2_{Sm}$ calculated from emission spectra (**b**).

The luminescent spectra for $2_{Sm}$ under varying $\lambda_{ex}$ are shown in Figure 7a. With an increase in the excitation wavelength from 340 nm to the higher wavelengths, a gradual red shift of phenanthroline emission band is observed, which may be due to the redistribution of the vibrational states of such a large conjugated system. Due to moving the main maximum of wide-banded emission from ca. 370 nm (UV region) to ca. 430 nm (visible region), the overall increase in blue color contribution is observed, as indicated by chromaticity diagram (Figure 7b). Further increasing the excitation wavelength leads to the weakening of blue emission bands, giving almost perfect white light at $\lambda_{ex}$ = 400 nm with the corresponding (x,y) coordinates of (0.323,0.327). The correlated color temperature (CCT), which is calculated for this point to be 5970 K, and 5% color purity for the 480 nm blue characteristic wavelength, indicate an almost perfect daylight color suitable

for the implementation in WLEDs. A further decrease in the excitation energy leads to the complete vanishing of blue emission, while a relatively weak emission of $Sm^{3+}$ is still retained even at $\lambda_{ex}$ = 480 nm, which further results in a gradual shift of the emission color to the yellow region upon blue light excitation. Such a complex nonlinear variation of $2_{Sm}$ luminescence color is apparently attributed to the abovementioned competing between different luminescence mechanisms, including ligand-centered emission of the coordinated phenanthroline and dipole–dipole energy transfer from phen antenna to the emissive $Sm^{3+}$ center. Although there are several reported examples of single-metal samarium MOF white emitters [71–74], no excitation-dependent color tuning was reported for Sm(III)-based carboxylate coordination compounds with any ligand containing 2,2′-bipyridyl cores, to the best of our knowledge. Therefore, $2_{Sm}$ is an unusual example of a pronounced tuning of the emission color in a monometallic lanthanide-based MOF, which makes it promising for obtaining multicolored luminophores and optical data processing.

## 3. Experimental

### 3.1. Materials

*Trans*-1,4-cyclohexanedicarboxylic acid ($H_2$chdc, >97.0%) and 1,10-phenanthroline monohydrate (phen·$H_2$O, >98.0%) were received from TCI. N,N-dimethylformamide (DMF, reagent grade) was received from Vekton. Cerium(III) nitrate hexahydrate (Ce(NO$_3$)$_3$·6H$_2$O, 99.5% REO) was received from Alfa Aesar. Lanthanide(III) chloride hydrates (LaCl$_3$·7H$_2$O, high-purity grade; PrCl$_3$·7H$_2$O, high-purity grade; NdCl$_3$·7H$_2$O, high purity grade; SmCl$_3$·6H$_2$O, reagent grade) were received from Novosibirsk Rare Metals Plant. All reagents were used as received without further purification.

### 3.2. Instruments

IR spectra in KBr pellets were recorded in the range $4000-400$ cm$^{-1}$ on a Bruker Scimitar FTS 2000 spectrometer. Elemental analysis was conducted with a VarioMICROcube analyzer. Powder X-ray diffraction (PXRD) analysis was performed at room temperature on a Bruker D8 Advance diffractometer (Cu-K$\alpha$ radiation, $\lambda$ = 1.54178 Å). Thermogravimetric analysis was carried out using a Netzsch TG 209 F1 Iris instrument under Ar flow (30 cm$^3$·min$^{-1}$) at a 10 K·min$^{-1}$ heating rate. Photoluminescence spectra were recorded with a spectrofluorometer Horiba Jobin Yvon Fluorolog 3 equipped with ozone-free Xe-lamp 450W power, cooled photon detector R928/1860 PFR technologies with refrigerated chamber PC177CE-010 and double-grating monochromators. The spectra were corrected for source intensity and detector spectral response by standard correction curves.

### 3.3. Synthetic Methods

Synthesis of **1**. Amounts of 55.0 mg (0.148 mmol) of LaCl$_3$·7H$_2$O, 30.0 mg (0.152 mmol) of phen·$H_2$O and 53.0 mg (0.308 mmol) of $H_2$chdc were mixed in a glass vial and dissolved in the mixture of 7.5 mL of DMF and 1.5 mL of water. Then, the obtained solution was closed by a screw cap and heated at 80 °C for 2 days. The formed white precipitate was filtered off, washed with DMF and dried in air. A single crystal suitable for SCXRD was taken directly from the mother liquor before filtration. Yield: 33.5 mg (44%). IR spectrum (KBr, cm$^{-1}$) main bands: 3441 (m, br, $\nu$O–H); 2928 (m, $\nu$Csp$^3$–H); 2857 (m, $\nu$Csp$^3$–H); 1674 (w, $\nu$CO$_{DMF}$); 1606, 1580 and 1551 (m, $\nu$COO$_{as}$); 1410 (m, $\nu$COO$_s$). Elemental analysis data for the sample **1′** after the filtration of **1** and its storage (%), calculated for La$_2$(C$_3$H$_7$NO)$_{1.5}$(H$_2$O)$_{0.5}$(C$_8$H$_{10}$O$_4$)$_3$: C, 37.7; H, 4.6; N, 2.3. Found: C, 37.9; H, 4.7; N, 2.5. TG data: 12% weight loss at 280 °C. Calculated for 1.5DMF+0.5H$_2$O: 13%.

Synthesis of **2$_{Ce}$**. Amounts of 43.0 mg (0.099 mmol) of Ce(NO$_3$)$_3$·6H$_2$O, 20.0 mg (0.101 mmol) of phen·$H_2$O and 26.0 mg (0.151 mmol) of $H_2$chdc were mixed in a glass vial and dissolved in the mixture of 1.2 mL of DMF and 0.8 mL of water. Then, the obtained solution was closed by a screw cap and heated at 80 °C for 2 days. The formed yellow precipitate was filtered off, washed with DMF and dried in air. A single crystal suitable for SCXRD was taken directly from the mother liquor before filtration. Yield:

16.3 mg (27%). IR spectrum (KBr, cm$^{-1}$) main bands: 3440 (w, br, $\nu$O–H); 3073 and 3059 (w, $\nu$Csp$^2$–H); 3012 and 2998 (w, $\nu$Csp$^2$–H); 2934 (m, $\nu$Csp$^3$–H); 2858 (m, $\nu$Csp$^3$–H); 1685 (s, $\nu$CO$_{DMF}$); 1586 (s, $\nu$COO$_{as}$); 1411 (s, $\nu$COO$_s$). Elemental analysis data (%), calculated for [Ce$_2$(C$_{12}$H$_8$N$_2$)$_2$(C$_8$H$_{10}$O$_4$)$_3$]·0.75C$_3$H$_7$NO·H$_2$O: C, 49.3; H, 4.4; N, 5.4. Found: C, 49.4; H, 4.5; N, 5.6. TG data: 5% weight loss until 400 °C. Calculated for 0.75DMF+H$_2$O: 6%.

Synthesis of **2$_{Pr}$**, **2$_{Nd}$** and **2$_{Sm}$** was carried out analogously to the synthesis of **2$_{Ce}$**, except for changing the cerium nitrate to the corresponding amounts of PrCl$_3$·7H$_2$O, NdCl$_3$·7H$_2$O or SmCl$_3$·6H$_2$O and reducing the reaction time to 20 h.

**2$_{Pr}$**. Yield of greenish precipitate: 15.3 mg (26%). IR spectrum (KBr, cm$^{-1}$) main bands: 3433 (w, br, $\nu$O–H); 3072 and 3058 (w, $\nu$Csp$^2$–H); 3009 and 2996 (w, $\nu$Csp$^2$–H); 2929 (m, $\nu$Csp$^3$–H); 2856 (m, $\nu$Csp$^3$–H); 1685 (s, $\nu$CO$_{DMF}$); 1586 (s, $\nu$COO$_{as}$); 1410 (s, $\nu$COO$_s$). Elemental analysis data (%), calculated for [Pr$_2$(C$_{12}$H$_8$N$_2$)$_2$(C$_8$H$_{10}$O$_4$)$_3$]·0.75C$_3$H$_7$NO·H$_2$O: C, 49.2; H, 4.4; N, 5.3. Found: C, 49.1; H, 4.4; N, 5.4. TG data: 4.5% weight loss until 430 °C. Calculated for 0.75DMF+H$_2$O: 6%.

**2$_{Nd}$**. Yield of pinkish-blue precipitate: 15.0 mg (25%). IR spectrum (KBr, cm$^{-1}$) main bands: 3429 (w, br, $\nu$O–H); 3074 and 3059 (w, $\nu$Csp$^2$–H); 3011 and 2998 (w, $\nu$Csp$^2$–H); 2929 (m, $\nu$Csp$^3$–H); 2856 (m, $\nu$Csp$^3$–H); 1685 (s, $\nu$CO$_{DMF}$); 1585 (s, $\nu$COO$_{as}$); 1410 (s, $\nu$COO$_s$). Elemental analysis data (%), calculated for [Nd$_2$(C$_{12}$H$_8$N$_2$)$_2$(C$_8$H$_{10}$O$_4$)$_3$]·0.75C$_3$H$_7$NO·0.5H$_2$O: C, 49.3; H, 4.3; N, 5.4. Found: C, 49.2; H, 4.4; N, 5.3. TG data: 4% weight loss until 440 °C. Calculated for 0.75DMF+0.5H$_2$O: 5%.

**2$_{Sm}$**. Yield of white precipitate: 17.4 mg (29%). IR spectrum (KBr, cm$^{-1}$) main bands: 3443 (w, br, $\nu$O–H); 3076 and 3060 (w, $\nu$Csp$^2$–H); 3011 and 2998 (w, $\nu$Csp$^2$–H); 2929 (m, $\nu$Csp$^3$–H); 2856 (m, $\nu$Csp$^3$–H); 1685 (s, $\nu$CO$_{DMF}$); 1588 (s, $\nu$COO$_{as}$); 1413 (s, $\nu$COO$_s$). Elemental analysis data (%), calculated for [Sm$_2$(C$_{12}$H$_8$N$_2$)$_2$(C$_8$H$_{10}$O$_4$)$_3$]·0.75C$_3$H$_7$NO: C, 49.2; H, 4.2; N, 5.4. Found: C, 49.0; H, 4.3; N, 5.3. TG data: 4% weight loss until 450 °C. Calculated for 0.75DMF: 4.5%.

Synthesis of **3**. Amounts of 38.0 mg (0.101 mmol) of NdCl$_3$·7H$_2$O, 20.0 mg (0.101 mmol) of phen·H$_2$O and 34.5 mg (0.201 mmol) of H$_2$chdc were mixed in a glass vial and dissolved in 5.0 mL of DMF. Then, the obtained solution was closed by a screw cap and heated at 110 °C for 4 h. For performing PXRD, the resulting white precipitate was filtered off, washed by DMF and dried in air. Due to a weak diffraction from the obtained single crystals, for performing SCXRD analysis, mother liquor was replaced by pure DMF and then crystals of **3** were carefully stored in this form until synchrotron working trip. Yield: less than 10%.

*3.4. Single-Crystal X-ray Diffraction Details*

Diffraction data for single crystals of **1** and **2$_{Ln}$** were collected on an automated Agilent Xcalibur diffractometer equipped with an area AtlasS2 detector (graphite monochromator, $\lambda$(MoK$\alpha$) = 0.71073 Å). Integration, absorption correction and determination of unit cell parameters were performed using the CrysAlisPro program package [75]. The structures were solved by the dual-space algorithm (SHELXT [76]) and refined by the full-matrix least-squares technique (SHELXL [77]) in the anisotropic approximation (except hydrogen atoms). Positions of hydrogen atoms of organic ligands were calculated geometrically and refined in the riding model. Diffraction data for single crystals of **3** were obtained on the 'Belok' beamline [78,79] ($\lambda$ = 0.745 Å) of the National Research Center 'Kurchatov Institute' (Moscow, Russian Federation) using a Rayonix SX165 CCD detector. The data were indexed, integrated and scaled, and absorption correction was applied using the XDS program package [80]. The crystallographic data and details of the structure refinements are summarized in Tables S1 and S2.

**4. Conclusions**

To summarize, six new metal–organic frameworks based on early lanthanide(III) cations and *trans*-1,4-cyclohexanrdicarboxylic acid (H$_2$chdc) were synthesized and structurally characterized. All the compounds possess three-dimensional coordination lattices.

However, a lanthanide contraction appears to provide a significant composition difference between lanthanum-based $[La_2(H_2O)_4(chdc)_3]$ with $CN(La^{3+}) = 10$ and lanthanide-based $[Ln_2(phen)_2(chdc)_3]$ frameworks with $CN(Ln^{3+}) = 9$, further resulting in a change in the main structural motif from one-dimensional metal–carboxylate chains to separate binuclear carboxylate building units. A variation of synthetic conditions gave two topological isomers of the $[Nd_2(phen)_2(chdc)_3]$ coordination framework, bearing primitive cubic **pcu** topology or rare hexagonal helical **snz** topologies. Luminescent measurements revealed a strong and nonmonotonous tuning of luminescence color for the compound $[Sm_2(phen)_2(chdc)_3]\cdot 0.75DMF$, while excitation wavelength variated, with an emission color change from pinkish red at $\lambda_{ex} = 340$ nm to white at $\lambda_{ex} = 400$ nm and then to yellow at lower excitation energies. Such behavior represents a rare example of both a white-light emitter and highly tunable luminophore based on a single lanthanide ion.

**Supplementary Materials:** The following supporting information can be downloaded at: https://www.mdpi.com/article/10.3390/inorganics10100163/s1, Table S1: Single-crystal XRD experiment and refinement details for **1**, **2$_{Ce}$**, **2$_{Pr}$**; Table S2: Single-crystal XRD experiment and refinement details for **2$_{Nd}$**, **2$_{Sm}$**, **3**; Figure S1: Experimental (black) and theoretical (red) PXRD patterns for **2$_{Ce}$**; Figure S2: Experimental (black) and theoretical (red) PXRD patterns for **2$_{Pr}$**; Figure S3: Experimental (black) and theoretical (red) PXRD patterns for **2$_{Nd}$**; Figure S4: Experimental (black) and theoretical (red) PXRD patterns for **2$_{Sm}$**; Figure S5: PXRD patterns for **2$_{Pr}$** temperature- and time-screening syntheses; Figure S6: PXRD patterns for **2$_{Sm}$** temperature- and time-screening syntheses; Figure S7: Coordination modes of $(k^2,k^1;k^2,k^1)$-chdc ligand (**a**) and $(k^2;k^2)$-chdc ligand (**b**); Figure S8: Space-filling representation of channels in the coordination framework of **1**; Figure S9: Experimental (black) PXRD pattern for **1′** and theoretical (red) one for **1** (a). View along *c* crystallographic axis in **1** (b); Figure S10: TGA pattern and relative dm/dT curve for the sample **1′**; Table S3: Selected bond lengths in **2$_{Ln}$** and **3**; Figure S11: Infrared spectra for **1′** and **2$_{Ln}$**; Table S4: Details on TG decomposition steps of **1′** and **2$_{Ln}$**; Figure S12: Excitation (red) spectrum for **2$_{Nd}$** at $\lambda_{em} = 1077$ nm and emission (black) spectrum for **2$_{Nd}$** at $\lambda_{ex} = 804$ nm; Figure S13: Excitation spectrum for **2$_{Sm}$** at $\lambda_{em} = 580$ nm.

**Author Contributions:** P.A.D.: Conceptualization, methodology, validation, investigation, writing—original draft preparation, visualization; A.A.R.: methodology, validation, investigation; V.P.F.—resources, writing—review and editing, project administration, funding acquisition. All authors have read and agreed to the published version of the manuscript.

**Funding:** This work was supported by the Ministry of Science and Higher Education of the Russian Federation (Agreement No. 075-15-2022-263). The experiments were performed using large-scale research facilities "EXAFS spectroscopy beamline". Analytical services were provided under projects No. 121031700321-3 and 121031700313-8.

**Data Availability Statement:** CCDC 2190414–2190419 contains supplementary crystallographic data for this paper. These data can be obtained free of charge from The Cambridge Crystallographic Data Center [81].

**Conflicts of Interest:** The authors declare no conflict of interest.

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
