# Peer review of "Aliphatic-Bridged Early Lanthanide Metal–Organic Frameworks: Topological Polymorphism and Excitation-Dependent Luminescence"

_inorganics, doi:10.3390/inorganics10100163_

Round 1

Reviewer 1 Report

This is an interesting paper that describes six new metal metal-organic frameworks (MOFs). In particular, the 2Sm compound shows an interesting complex nonlinear variation of luminescence color, which is worth to be reported.  I do not have technical questions or objections and I recommend the manuscript acceptance as is.

Author Response

Authors thank the reviewer for such kindly high evaluation of the work.

Reviewer 2 Report

The authors synthesized 6 new MOFs based on early lanthanide(III) cations and H2chdc, among which 2Sm would change its emission color while excitation wavelength variated. This highly tunable luminophore based on single lanthanide ion has potential to be applied in light-emitting devices and optical data processing. I therefore recommend its publication in Inorganics after some revisions:

1.       The introduction of lanthanide MOF for white light emitting materials should be provided.

2.       Please provide the quantum yield of 2Sm.

3.       Please provide the images of 2Sm under different excitation wavelength, because it is potential for WLED.

4.       In all the figures, there should be "/" between the viable and unit, such as "T / °C", instead of ",".

5.       In Figure 3-5, the symbol of crystal axis should be italic, i.e., the "a" and "b" should be italic.

6.       In Figure 7b, the numbers in CIE diagram axis should be "0.0", not "0,0".

Author Response

The authors synthesized 6 new MOFs based on early lanthanide(III) cations and H2chdc, among which 2Sm would change its emission color while excitation wavelength variated. This highly tunable luminophore based on single lanthanide ion has potential to be applied in light-emitting devices and optical data processing. I therefore recommend its publication in Inorganics after some revisions:

Authors are grateful to the reviewer for high evaluation of the work.

  1. The introduction of lanthanide MOF for white light emitting materials should be provided.

The introduction is now appended by the part concerning whit emitting MOFs:

In particular, combining lanthanide(III) ions bearing different emission colors and/or using highly emissive organic guests or ligands within Ln3+-based MOFs is a convenient route for the synthesis of single phase white light emitters suitable for real applications [24–29].

  1. Please provide the quantum yield of 2Sm.
  2. Please provide the images of 2Sm under different excitation wavelength, because it is potential for WLED.

[Sm2(phen)2(chdc)3]·0.75DMF (2Sm) represents a rare example of monometallic Sm-based luminophore with highly tunable color. An excitation dependence of emission color is based on the competing between the emission of large aromatic π-system (i.e. phenanthroline) and Sm3+ cation, which is known to be mildly emissive. Unfortunately, attempts to determine QY for 2Sm failed due to the low intensity of its emission. However, appropriate changes in emission color are easily observed visually, as shown on digital photographs (see figure in the attached PDF), confirming the spectral data.

  1. In all the figures, there should be "/" between the viable and unit, such as "T / °C", instead of ",".
  2. In Figure 3-5, the symbol of crystal axis should be italic, i.e., the "a" and "b" should be italic.
  3. In Figure 7b, the numbers in CIE diagram axis should be "0.0", not "0,0".

Authors thank the reviewer for important corrections. Figures have been revised according to these notes.

Reviewer 3 Report

This article reports crystal structures, thermal behavior, and luminescence properties of six new three-dimensional metal-organic frameworks based on early lanthanide(III) cations and trans-1,4-cyclohexanedicarboxylic acid.

From my point of view, the manuscript is interesting and will attract the interest of many scientists working with such systems. However, there are a few issues that the authors are kindly requested to take into account for a revised version.

1. Almost all of the drawings (Figs. 2, 6, 7, S1-10 attached to the manuscript or supporting information require improvement as they were made extremely carelessly. All the figures mentioned should have a Y axis and should be legible. In Fig. 6 there should be a separate Y axis for the DTG curve.

2. In part 2.3 the authors claimed: "...the presence of weak HOH deformation bands in 2390…2310 cm–1 area indicates a water coordination to metal center, fully matching crystal structure data". However, there should also be additional bands in the range of 900–600 cm-1 attributed to coordination water molecules (K. Nakamoto, Infrared and Raman Spectra of Inorganic and Coordination Compounds). The authors are asked for careful analysis of FTIR spectra in this field.

3. The details of the thermal decomposition stages (temperature range, mass loss, and peak temperature) should be presented in table. The authors did not mention what was the final products of thermal decomposition of the inestigated materials, as well as what atmosphere was used. What is more, I suggest perfomed simultaneus DSC analysis. It facilitates the interpretation of decomposition steps.

4.  Part "3. Experimental" is absolutely incompleted. Although the synthesis was described there is no information on details of the methods used (X-ray, thermogravimetry, elemental analysis, FTIR, emission spectra), i.e. apparatus used, measurement conditions.

If these changes are made, the paper will be suitable for publication in "Inorganics".

Author Response

This article reports crystal structures, thermal behavior, and luminescence properties of six new three-dimensional metal-organic frameworks based on early lanthanide(III) cations and trans-1,4-cyclohexanedicarboxylic acid.

From my point of view, the manuscript is interesting and will attract the interest of many scientists working with such systems. However, there are a few issues that the authors are kindly requested to take into account for a revised version.

Authors are grateful to the reviewer for such kindly high evaluation of the work and many fair-minded notes which helped to improve the article.

  1. Almost all of the drawings (Figs. 2, 6, 7, S1-10 attached to the manuscript or supporting information require improvement as they were made extremely carelessly. All the figures mentioned should have a Y axis and should be legible. In Fig. 6 there should be a separate Y axis for the DTG curve.

Y axes are added now. Single figures in the MS have been enlarged to improve their readability.  

  1. In part 2.3 the authors claimed: "...the presence of weak HOH deformation bands in 2390…2310 cm–1 area indicates a water coordination to metal center, fully matching crystal structure data". However, there should also be additional bands in the range of 900–600 cm-1 attributed to coordination water molecules (K. Nakamoto, Infrared and Raman Spectra of Inorganic and Coordination Compounds). The authors are asked for careful analysis of FTIR spectra in this field.

Authors thank the reviewer for the comment. A description of 1’ is now transferred to ESI (pages 10-12), its IR spectrum has been reinterpreted:

IR spectrum of 1′ (Fig. S11) contains characteristic absorption bands for water O–H vibrations, cyclohexane ring and DMF methyl group C(sp3)–H vibrations, DMF C=O vibrations and COO group antisymmetric and symmetric stretchings. The absence of aromatic C(sp2)–H absorption bands in 3075-3060 cm–1 region, as well as the absence of strong bands at ca. 1583, 1534 and 850 cm–1 confirm the absence of phenanthroline in 1’ in any (coordinated or guest) form, fully matching crystal structure data obtained for its parent compound 1.

The interpretation of 2Ln IR spectra in the main text has also been revised:

A presence of weak coupled bands at 3075-3060 cm–1 corresponding to the valence vibrations of C(sp2)–H bonds, as well as strong bands at ca. 1583, 1534 and 850 cm–1, belonging to the phenanthroline oscillations [50,51], prove the successful incorporation of phen ligand into coordination frameworks 2Ln instead of 1, fully consisting with their crystal structures.

  1. The details of the thermal decomposition stages (temperature range, mass loss, and peak temperature) should be presented in table. The authors did not mention what was the final products of thermal decomposition of the inestigated materials, as well as what atmosphere was used. What is more, I suggest perfomed simultaneus DSC analysis. It facilitates the interpretation of decomposition steps.

The presented table has been added to the ESI file as table S4. Weights of final residues at 600 °C show that final metal oxides are apparently contaminated by a significant amount of carbon species, which are not fully oxidized and evaporated due to performing TG analysis under Ar inert atmosphere. The corresponding text has been added to the discussion:

Weights of final residues at 600 °C exceed the calculated weights of lanthanide oxides (table S4), what indicates a presence of considerable amount of carbon admixture in the thermolysis products of 2Ln, due to incomplete evaporation of organic moieties.

Thermogravimetric analysis (TGA) allows to verify guest content in porous metal-organic framework samples as well as to characterize a thermal stability of the coordination lattice. A detailed investigation of decomposition steps, which are closely situated in ca. 450…530 °C range for 2Ln, by performing DSC may be helpful and possible in future, but confidently falls away from the scope of this work.   

  1. Part "3. Experimental" is absolutely incompleted. Although the synthesis was described there is no information on details of the methods used (X-ray, thermogravimetry, elemental analysis, FTIR, emission spectra), i.e. apparatus used, measurement conditions.

Following the reviewer’s kind suggestion, this information has been transferred into the manuscript from ESI. Please check parts 3.1, 3.2 and 3.4 in the revised version of MS.

Reviewer 4 Report

In this paper, three series of six new metal-organic frameworks based on lanthanide(III) cations and trans-1,4-cyclohexanrdicarboxylic acid (H2chdc) were synthesized and structurally characterized. Luminescent measurements showed that compound [Sm2(phen)2(chdc)3]·0.75DMF has a strong and non-monotonous tuning of luminescence color. Some content descriptions in the article are unclear, and some experiments need to be further supplemented. Major revisions are recommended.

1.     In introduction, rare earth complexes are often insoluble in water, which is not conducive to biological imaging. Please modify the relevant content and literatures.

2.     In 2.1. part, “The role of 1,10-phenathroline in the synthesis could be therefore rationalized in terms of its slight basicity promoting the deprotonation of weak organic acid H2chdc.” There is no experimental and literature support for this statement.

3.     I don't understand why the authors put compound 1' in the article? From powder diffraction, the matching degree of 1 and 1' is relatively small, and it is difficult to see the similarity of the structure from the data, so it is impossible to use 1' for property experiments. There is also no property test for 1' in the article. In addition, from powder diffraction and infrared images, the peak angles and wavenumber positions of 1' and series 2 compounds are also very similar. Considering the radius and proximity of rare earth ions, is it possible that 1' is an analog of series 2 compounds? If the author needs to put 1' in the article for discussion, please add relevant proof experiments.

4.     Difference between La-based 1 and series 2Ln is not suggested to be explained from lanthanide contraction. Because, the radius of lanthanide shrinkage reduction is very small. In addition, in series 2Ln, 1,10-phen participates in the coordination. The volume of 1,10-phen is much larger than that of water, and the steric hindrance will also increase during coordination. It is reasonable that water is more likely to participate in the coordination.

5.     In 2.2 part, “The asymmetric unit of 1 includes one La atom.” This sentence is inappropriately described. It is suggested that this sentence “four O atoms of two κ2,κ1- carboxylic groups, two O atoms of two more κ2,κ 1- carboxylic groups” and “two O atoms of κ2,κ1- carboxylic group, one O atom of one more κ2,κ1- carboxylic group” be changed to “six O atoms of two κ2,κ1- carboxylic groups,” and “three O atoms of two κ2,κ1- carboxylic groups,”, respectively. From Figure 3c, the 010 Miller index should be the 100 index. Please supplement the coordination environment picture of the ligand H2chdc and discuss it separately. Please mark the exact location of the measured aperture on the diagram. Please supplement the reference documentation of the software used.

6.     In 2.3 part, the series 2Ln compounds contain solvent molecules, but no weight loss occurs until the skeleton begins to collapse at 500 degrees Celsius, please explain.

7.     In 2.4 part, please explain what the ligand is in “apparently attributed to the ligand-centered”. From Figure 7a, phenanthroline emission peak has been red-shifted, but it is described as blue-shifted in the text, please explain.

Author Response

In this paper, three series of six new metal-organic frameworks based on lanthanide(III) cations and trans-1,4-cyclohexanrdicarboxylic acid (H2chdc) were synthesized and structurally characterized. Luminescent measurements showed that compound [Sm2(phen)2(chdc)3]·0.75DMF has a strong and non-monotonous tuning of luminescence color. Some content descriptions in the article are unclear, and some experiments need to be further supplemented. Major revisions are recommended.

Authors thank the reviewer for the careful evaluation of the manuscript. We have addressed all the reviewer’s notes.

  1. In introduction, rare earth complexes are often insoluble in water, which is not conducive to biological imaging. Please modify the relevant content and literatures.

The reference list is updated now by new references [36-40].

  1. In 2.1. part, “The role of 1,10-phenathroline in the synthesis could be therefore rationalized in terms of its slight basicity promoting the deprotonation of weak organic acid H2chdc.” There is no experimental and literature support for this statement.

The word “rationalized” was replaced by “suggested” according to the reviewer’s fair note. New references [42,43], revealing the basic nature of phenanthroline, were added to support this suggestion. 

  1. I don't understand why the authors put compound 1' in the article? From powder diffraction, the matching degree of 1 and 1' is relatively small, and it is difficult to see the similarity of the structure from the data, so it is impossible to use 1' for property experiments. There is also no property test for 1' in the article. In addition, from powder diffraction and infrared images, the peak angles and wavenumber positions of 1' and series 2 compounds are also very similar. Considering the radius and proximity of rare earth ions, is it possible that 1' is an analog of series 2 compounds? If the author needs to put 1' in the article for discussion, please add relevant proof experiments.

A comparison of 1’ experimental diffractogram and 2Ce theoretical diffractogram is shown below (see figure in the attached PDF) and demonstrates almost no peak matching. Infrared spectra (Fig. S11) also show the absence of mostly characteristic phenanthroline bands (3075-3060, 1583, 1534 and 850 cm–1) in 1’, which present in 2Ln spectra. Therefore, no phen incorporation is confidently assumed for 1’.

Synthesis of compound 1 single crystals is fully reproducible and it was checked several times by unit cell parameters determination for selected crystals. Apparently, 1 is not stable after filtration and undergoes a transition to the solid denoted as 1’. Although the crystal structure of 1’ is not determined, conclusions concerning its elemental, molecular composition and thermal behavior, on the basis of CHN, IR, TG and X-ray structure of its origin 1, are decently reasonable. Indeed, luminescent properties of MOFs depend strongly on their local structural features, so the luminescence study for 1’ would be incorrect and therefore it has not been performed.

We agree to the reviewer that discussion concerning possible molecular mobility in the coordination framework of 1 seems to be not fully reliable and therefore we decided to leave only 1 crystal structure discussion in the manuscript, transferring all the descriptions concerning 1’ from the discussion part to the ESI file (pages 10-12).

  1. Difference between La-based 1 and series 2Ln is not suggested to be explained from lanthanide contraction. Because, the radius of lanthanide shrinkage reduction is very small. In addition, in series 2Ln, 1,10-phen participates in the coordination. The volume of 1,10-phen is much larger than that of water, and the steric hindrance will also increase during coordination. It is reasonable that water is more likely to participate in the coordination.

We agree to the reviewer that coordination of more sterically hindered phenanthroline instead of water, while ionic radius of Ln3+ decreases, seems quite surprising. However, a water substitution by phen in the structure of 2Ce is accompanied by the change in the coordination mode of carboxylate groups compared to 1La, what results in the decrease of metal coordination number from 10 (La) to 9 (Ce and further). Such coordination number reduction apparently compensates the larger molecular size of phen. There are several other reported examples of the impact of lanthanide contraction on the coordination framework topologies [new references 44-46 in the manuscript].

  1. In 2.2 part, “The asymmetric unit of 1 includes one La atom.” This sentence is inappropriately described. It is suggested that this sentence “four O atoms of two κ2,κ1- carboxylic groups, two O atoms of two more κ2,κ 1- carboxylic groups” and “two O atoms of κ2,κ1- carboxylic group, one O atom of one more κ2,κ1- carboxylic group” be changed to “six O atoms of two κ2,κ1- carboxylic groups,” and “three O atoms of two κ2,κ1- carboxylic groups,”, respectively.

Authors thank the reviewer for pointing such inaccuracy in text. The corresponding fragments have been changed tofour O atoms of two chelated κ2,κ1- carboxylic groups, two O atoms of two non-chelated κ2,κ 1- carboxylic groups” and “two O atoms of chelated κ2,κ1- carboxylic group, one O atom of non-chelated κ2,κ1- carboxylic group”, respectively.

From Figure 3c, the 010 Miller index should be the 100 index.

Mercury software was used for 1 theoretical diffractogram indexing and hkl planes visualization. Intermetal La…La distances, corresponding to (κ22) and (κ2121) chdc bridges, are ca. 11.77 Å and 10.54 Å, respectively. They indeed slightly differ from (0 0 1) and (0 1 0) interplanar distances possessing values of 11.00 Å and 10.06 Å, respectively, due to non-perpendicularity of crystallographic axes in the triclinic cell. However, the mentioned ligands have a major contribution into these interplanar distances. (–1 1 0) index interplanar distance corresponds better to (κ2121)-chdc bridge than (0 1 0) index interplanar distance, but low intensity of  (–1 1 0) reflection in the theoretical diffractogram makes impossible any convenient qualitative analysis of its differences between 1 and 1’ PXRD patterns.

Please supplement the coordination environment picture of the ligand H2chdc and discuss it separately.

Coordination modes of two types of chdc ligands are now shown in Figure S1.

Please mark the exact location of the measured aperture on the diagram.

Figure S2 now shows a space-filling representation of channels in highly porous coordination framework 1.

Please supplement the reference documentation of the software used.

Diamond 3.0b was used for the construction of all crystal structure figures and for the estimation of channel apertures.

PLATON 10 was used for the calculation of void volumes in the coordination frameworks and validation of the structures.

Cambridge Structural Database (CSD, May 2022 update) was used for the structure searches. Mercury 2020.3.0 included in CSD software pack was used for the construction of theoretical PXRD patterns for all the structures and for indexation of theoretical PXRD pattern for 1.

ChemDraw 18.0 was used for the construction of Figure 1 in the manuscript.

Origin 9.6 was used for the numerical data processing and visualization (i.e. IR, TGA, PXRD, luminescence data)

Software for crystal data acquisition, integration, structure solution and refinement is referenced in the experimental part.

  1. In 2.3 part, the series 2Ln compounds contain solvent molecules, but no weight loss occurs until the skeleton begins to collapse at 500 degrees Celsius, please explain.

Due to high molar masses of 2Ln and low guest content, the observed several percent TG weight losses below ~450 °C are reasonable. Please check the interpretation of CHN and TG data for each compound in part 3.3 and the scaled part of thermograms shown in figure presented below in the attached PDF. Minor additional weight loss (1-2 %) for 2Ce below 250 °C apparently results from its higher degree of hydration (possibly on the surface of crystallites), which is also observed by CHN analysis (part 3.3).

  1. In 2.4 part, please explain what the ligand is in “apparently attributed to the ligand-centered”. From Figure 7a, phenanthroline emission peak has been red-shifted, but it is described as blue-shifted in the text, please explain.

Authors thank the reviewer for such careful observation. A corresponding fragment was corrected to “… apparently attributed to phenanthroline ligand-centered …”

The text concerning moving the eye-visible emission color to the blue region while increasing (red-shifting) of λem maxima has been rewritten more accurately:

With an increase in the excitation wavelength from 340 nm to the higher wavelengths, a gradual red shift of phenanthroline emission band is observed, which may be due to the redistribution of the vibrational states of such a large conjugated system. Due to moving the main maximum of wide-banded emission from ca. 370 nm (UV region) to ca. 430 nm (visible region), the overall increase of blue color contribution is observed, as indicated by chromaticity diagram (fig. 7b). Further increasing the excitation wavelength leads to the weakening of blue emission bands giving almost perfect white light (fig. 7b) at λex = 400 nm with the corresponding (x,y) coordinates of (0.323,0.327).

Round 2

Reviewer 3 Report

The authors provided sufficient improvement of the manuscript and now it is suitable for publication in Inorganics.

Reviewer 4 Report

Accept in present form.